

# Prevalence of bacterial vaginosis in Portuguese pregnant women and vaginal colonization by *Gardnerella vaginalis*

Daniela Machado[1], Joana Castro[1,2], José Martinez-de-Oliveira[3,4], Cristina Nogueira-Silva[5,6,7] and Nuno Cerca[1]

[1] Centre of Biological Engineering (CEB), Laboratory of Research in Biofilms Rosário Oliveira (LIBRO), University of Minho, Braga, Portugal
[2] Instituto de Ciências Biomédicas Abel Salazar, University of Porto, Porto, Portugal
[3] Women & Child Health Department, Centro Hospitalar Cova da Beira EPE, Covilhã, Portugal
[4] CICS-UBI, Health Sciences Research Center, Faculty of Health Sciences, University of Beira Interior, Covilhã, Portugal
[5] Department of Obstetrics and Gynecology, Hospital de Braga, Braga, Portugal
[6] Life and Health Sciences Research Institute, School of Medicine, University of Minho, Braga, Portugal
[7] ICVS/3B's, Braga/Guimarães, Portugal

Corresponding author
Nuno Cerca,
nunocerca@ceb.uminho.pt

## ABSTRACT

**Background.** We aimed to determine the prevalence of vaginal colonization by *Gardnerella vaginalis* and of bacterial vaginosis (BV) in Portuguese pregnant women, and to identify risk factors for BV and *G. vaginalis* colonization in pregnancy.

**Methods.** A cross-sectional study was conducted among pregnant women aged $\geq 18$ years who were attending in two public hospitals of the Northwest region of Portugal. Epidemiological data was collected by anonymous questionnaire. BV was diagnosed by Nugent criteria and *G. vaginalis* presence was identified by polymerase chain reaction. Crude associations between the study variables and BV or *G. vaginalis* colonization were quantified by odds ratios (ORs) and their 95% confidence intervals (CIs).

**Results.** The prevalences of BV and of *G. vaginalis* colonization among Portuguese pregnant women were 3.88% and 67.48%, respectively. Previous preterm delivery and colonization by *G. vaginalis* were factors with very high OR, but only statistically significant for a 90% CI. Conversely, higher rates of *G. vaginalis* colonization were found in women with basic educational level (OR = 2.77, 95% CI [1.33–5.78]), during the second trimester of pregnancy (OR = 6.12, 95% CI [1.80–20.85]) and with BV flora (OR = 8.73, 95% CI [0.50–153.60]).

**Discussion.** Despite the lower number of women with BV, prevalence ratios and association with risk factors were similar to recent European studies. However, the percentage of healthy women colonized by *G. vaginalis* was significantly higher than many previous studies, confirming that *G. vaginalis* colonization does not always lead to BV development.

## INTRODUCTION

Worldwide, bacterial vaginosis (BV) is considered to be one of the most prevalent gyneco-logical disorders of reproductive-age women and one of the most common causes of vaginal symptoms, prompting women to seek medical care (*Sobel, 1997*). This vaginal infection is microbiologically characterized by the replacement of a *Lactobacillus*-dominated vaginal microbiota by variable mixtures of strictly and facultative anaerobic bacteria including *Gardnerella vaginalis, Atopobium vaginae* and *Mobiluncus* spp. (*Verhelst et al., 2004*).

Currently, the presence of BV during pregnancy has attracted the attention of both clinicians and the scientific community, due to its relation with adverse pregnancy consequences such as preterm delivery (*Leitich et al., 2003*) and miscarriage (*Leitich & Kiss, 2007*). The mechanisms involved in pregnancy complications induced by BV are not fully clarified, but many researchers suggest that preterm delivery results from bacteria ascension from vagina to the membranes and amniotic fluid (*Goldenberg, Hauth & Andrews, 2000*; *Pararas, Skevaki & Kafetzis, 2006*). In recent years, several studies demonstrated that the presence or high load of vaginal microbes associated to BV such as *G. vaginalis*, *A. vaginae*, *Mobiluncus* spp., *Mycoplasma* spp. and *Leptotrichia/Sneathia* species, were related with an increased risk of preterm birth (*Foxman et al., 2014*; *Nelson et al., 2014*; *Bretelle et al., 2015*; *Kuon et al., 2017*). However, the great diversity of vaginal microbiota in women with BV affects its potential to develop preterm birth and other negative outcomes in pregnancy (*Hyman et al., 2014*; *Nelson et al., 2015*) and it becomes BV treatment in pregnant women a challenging task (*Haahr et al., 2016*). Despite of its high impact, few epidemiological studies on BV have been conducted in Portugal (*Guerreiro, Gigante & Teles, 1998*; *Silva et al., 2014*; *Machado et al., 2015*). Specifically, there is no information related to the prevalence of BV among Portuguese pregnant women. As such, we set this first epidemiological study aimed to determine the prevalences of BV and of *G. vaginalis* vaginal colonization in Portuguese pregnant women.

## MATERIALS & METHODS

### Population, data and biological samples collection

This cross-sectional study involved pregnant women attending prenatal care consultations in two public Portuguese hospitals of the Northwest region (Hospital of Braga and Unidade Local de Saúde de Matosinhos), between May 2014 and May 2016. The inclusion criteria for the present study were: to be pregnant at any gestational age and to be older than 18 years. The exclusion criteria were: women who were unable to read or understand Portuguese and who did not accept to participate in study.

During prenatal care consultation, obstetricians invited all eligible women under their care to participate in the study. All participants signed the informed consent form, as approved by the respective local ethical committees. Afterwards, the participants were questioned by the obstetricians about their age, educational level (basic, secondary or university), if they had previous BV episodes, previous deliveries, history of preterm delivery, suffered of chronic disease or smoking and about its use of intimate hygienic products and vitamin supplements, and determined the current gestational period (first,

second or third trimester). A short standardized questionnaire was used to collect these data. Finally, the obstetricians collected vaginal samples (only one sample point was taken for each participant), using sterile swab containing Amies transport medium with coal (VWR, Radnor, PA, USA). After collection, the swabs were immediately conserved at 4 °C and transported to the Laboratory of Research in Biofilms Rosário Oliveira of the University of Minho, where they were processed.

## Gram staining and BV diagnosis

For BV diagnosis, a direct smear was performed by transferring the vaginal fluid present on the swab to a glass slide. Then, vaginal smears were heat-fixed and Gram stained (*Spiegel, Amsel & Holmes, 1983*). Afterwards, the smears were visualized using an Olympus BX51 microscope (Olympus Portugal SA, Lisboa, Portugal) under oil immersion objective (1,000× magnification) and, subsequently, graded in accordance with the Nugent scoring system (*Nugent, Krohn & Hillier, 1991*). In summary, on 10 microscopic fields, the following bacterial morphotypes were identified and quantified: large gram-positive rods (*Lactobacillus* spp.), small gram-variable rods (*G. vaginalis*), small gram-negative rods (*Bacteroides* spp.) and curved gram-variable rods (*Mobiluncus* spp.). The sum of each morphotype score allowed to classify vaginal flora into normal (score 0–3), intermediate (score 4–6) or BV (score 7–10).

## Molecular detection of *G. vaginalis*

Each vaginal swab was then immersed in 2 mL of 0.9% (wt/v) of sodium chloride (Liofilchem; Roseto degli Abruzzi, Italy) and the content was suspended using vigorous vortexing. Afterwards, 1.5 mL of each diluted swab content was transferred into a tube that was centrifuged at 8,000 rpm, during 10 s, in order to allow for coal deposition at the bottom of the tube. Then, 0.6 mL of supernatant was collected to a new tube and this later was centrifuged at 13,500 rpm, during 5 min. At the end, the supernatant was discarded and the pellet was suspended in 0.1 mL of sterile ultra-pure water. This bacterial suspension was incubated in a heating block at 95 °C for 20 min in order to disrupt bacterial cell wall and release the cell content, making available the genomic DNA that is required for polymerase chain reaction (PCR). After incubation, it was immediately cooled on ice for 5 min and the cell suspension was centrifuged at 13,500 rpm, during 5 min. The supernatant was used as template for a multiplex PCR since we used the PCR to amplify different DNA sequences simultaneously (namely gene encoding 16S rRNA of *G. vaginalis* and *aap* gene of *Staphylococcus epidermidis*). Briefly, each PCR included 6 μL of Dream Taq PCR Master Mix 2X (Thermo Fisher Scientific, Waltham, MA, USA), 1 μL of DNA from the vaginal sample, 2 μL of distilled water DNase/RNAse free (Thermo Fisher Scientific), 0.5 μL of 5 μM forward (FW1) *G. vaginalis* primer , 0.5 μL of 5 μM reverse (RV1) *G. vaginalis* primer, 0.5 μL of 5 μM FW *S. epidermidis* primer, 0.5 μL of 5 μM RV *S. epidermidis* primer and 1 μL of complementary DNA of *S. epidermidis*. Also, a negative control (containing 1 μL of DNase/RNase free water instead of genomic DNA) and positive control (containing 1 μL of genomic DNA from pure culture of *G. vaginalis*) were included in each set of reactions. The tubes were placed in a thermocycler (Bio-Rad; Hercules, CA,
**Table 1 Primers used in this study.**

| Target | Set | Sequence (5′–3′) | TM (° C) | Amplification region (GenBank) | Reference |
|---|---|---|---|---|---|
| *G. vaginalis* 16SRNA | FW1 | CTCTTGGAAACGGGTGGTAA | 60 | KP996686.1 (from 99 to 399) | *Henriques et al. (2012)* |
| | RV1 | TTGCTCCCAATCAAAAGCGGT | 62 | | |
| | FW2 | AGCCTAGGTGGGCCATTACC | 59 | KP996686.1 (from 206 to 373) | *Castro et al. (2017)* |
| | RV2 | TGAGTAATGCGTGACCAACC | 55 | | |
| *S. epidermidis* aap | FW | GCACCAGCTGTTGTTGTACC | 59 | CP020463.1 (from 11,0863 to 11,1053) | *França et al. (2012)* |
| | RV | GCATGCCTGCTGATAGTTCA | 60 | | |

**Notes.**
TM, melting temperature; FW, forward; RV, reverse.

USA) that was programmed with the following protocol: 94 °C for 2 min, 40 steps of 94 °C for 30 s, 60 °C for 30 s, 72 °C for 1 min and finally 72 °C for 5 min. The inclusion of the specific primers for the *S. epidermidis aap* gene acted as internal positive control of PCR procedure, in order to ensure that amplifiable DNA was successfully extracted and there are no PCR inhibitors on sample.

After the PCR reaction, amplified products were analyzed in 1% (wt/v) of agarose (SeaKem LE; Rockland, ME, USA) gel with 0.05 µL/mL midori green nucleic acid dye (Nippon Genetics Europe GmbH, Düren, Germany). The electrophoresis run for 50 min at 100 volts. Finally, the results were visualized using the ChemiDoc (Bio-Rad) system, according to the manufacturer's instructions. Negative results for the amplification of the gene encoding 16S rRNA of *G. vaginalis* were confirmed using an independent set of primers (FW2/RV2). All primers used are described in Table 1 and were previously assessed for specificity (*França et al., 2012*; *Henriques et al., 2012*; *Castro et al., 2017*).

## Statistical analysis

Data were analyzed with GraphPad Prism version 6 (GraphPad Software Inc., La Jolla, CA, USA). Firstly, we determined BV and *G. vaginalis* colonization prevalences and then we used Chi-square or Fisher's exact test to verify whether BV positive status or *G. vaginalis* colonization were associated with some sociodemographic, medical, reproductive, behavioral or microbiological variables. A $p$-value $<0.05$ was used as threshold for statistically significance. The strength of association between BV diagnosis or *G. vaginalis* colonization with the study variables was assessed through calculation of odds ratios (OR) and their 95% confidence intervals (CI).

## Ethical considerations

The study was approved by the ethics committees of Unidade Local de Saúde de Matosinhos (process 013/CE/JAS) and of Hospital of Braga (process SECVS 063/2014). All the study participants agreed through informed consent to collaborate voluntarily, anonymously and freely. To ensure confidentiality no personal data were recorded that could lead to identification of the participants.
**Table 2  Characteristics of the studied population ($n = 206$).**

| Variables | (%) |
|---|---|
| Sociodemographic | |
|   Age (mean ± SD, years) | 30.00 ± 5.16 |
|     <30 years | 41.75 |
|     ≥30 years | 58.25 |
| Educational level | |
|   ≤Basic | 29.13 |
|   Secondary | 35.44 |
|   ≥University | 35.44 |
| Medical | |
|   Previous BV | 7.77 |
|   History of chronic disease | 14.56 |
| Reproductive | |
|   Previous delivery | 41.75 |
|   Previous preterm delivery | 6.80 |
|   Pregnancy trimester | |
|     First | 11.65 |
|     Second | 16.50 |
|     Third | 71.84 |
| Behavioral | |
|   Tobacco consumption | 12.62 |
|   Use of intimate hygiene products | 28.64 |
|   Vitamin supplementation | 86.41 |

**Notes.**

SD, standard deviation; BV, bacterial vaginosis.
Values are given as mean ± SD or percentage (%).

# RESULTS

Between May 2014 and May 2016, a total of 273 women followed in prenatal care consultation of two public Portuguese hospitals of the Northwest region agreed to participate in the current study. Among them, 67 had uninterpretable Gram-staining slides and where excluded from this study. Table 2 summarizes sociodemographic, medical, reproductive and behavioral characteristics of the studied population ($n = 206$). The participants had ages comprised between 19 and 41 years, resulting in mean age of 30.00 ± 5.16 years. Moreover, the majority of participants was in the third trimester of pregnancy (71.84%) and used vitamin supplementation (86.41%).

Among the 206 pregnant women, BV was diagnosed in only eight participants, resulting in a BV prevalence of 3.88% (Table 3). Probably due to the low number of BV cases, we did not found any significant statistically association between BV and the risk factors considered in this study ($p$-value > 0.05). Nevertheless, previous preterm delivery and colonization by *G. vaginalis* were factors with very high OR, that could be considered statistically significant for a 90% CI.

**Table 3  Characterization of women with or without BV.** Sociodemographic, medical, reproductive, behavioral and microbiological variables among women with or without BV.

| Variables | BV positive ($n = 8$) | BV negative ($n = 198$) | $p$ value | OR | 95% CI |
|---|---|---|---|---|---|
| **Sociodemographic** | | | | | |
| Age (mean ± SD, years) | 32.00 ± 3.16 | 29.91 ± 5.21 | 0.14 | | |
| <30 years | 1 | 85 | | 0.19 | 0.02–1.57 |
| ≥30 years | 7 | 113 | | 5.27 | 0.64–43.63 |
| Educational level | | | 0.97 | | |
| ≤Basic | 2 | 58 | | 0.81 | 0.16–4.11 |
| Secondary | 3 | 70 | | 1.10 | 0.26–4.73 |
| ≥University | 3 | 70 | | 1.10 | 0.26–4.73 |
| **Medical** | | | | | |
| Previous BV | 2 | 14 | 0.12 | 4.38 | 0.81–23.75 |
| History of chronic disease | 3 | 27 | 0.09 | 3.80 | 0.86–16.83 |
| **Reproductive** | | | | | |
| Previous delivery | 4 | 82 | 0.72 | 1.42 | 0.34–5.82 |
| Previous preterm delivery | 2 | 12 | 0.09 | 5.17 | 0.94–28.39 |
| Pregnancy trimester | | | 0.27 | | |
| First | 2 | 22 | | 2.67 | 0.51–14.04 |
| Second | 0 | 34 | | 0.28 | 0.02–4.98 |
| Third | 6 | 142 | | 1.18 | 0.23–6.04 |
| **Behavioral** | | | | | |
| Tobacco consumption | 0 | 26 | 0.60 | 0.38 | 0.02–6.84 |
| Use of intimate hygiene products | 4 | 55 | 0.23 | 2.60 | 0.63–10.76 |
| Vitamin supplementation | 7 | 171 | 1.00 | 1.11 | 0.13–9.35 |
| **Microbiological** | | | | | |
| *G. vaginalis* presence | 8 | 131 | 0.06 | 8.73 | 0.50–153.60 |

**Notes.**

BV, bacterial vaginosis; OR, odds ratio; CI, confidence interval; SD, standard deviation.

Values are given as mean ± SD or number.

Vaginal colonization by *G. vaginalis* was detected in 139 samples, representing a prevalence rate of 67.48% (Table 4). Statistically significant differences between *G. vaginalis* positive and negative groups were found in relation to maternal educational level, current pregnancy trimester and vaginal microflora profile ($p$ value < 0.05). Of note is that basic educational level (OR = 2.77, 95% CI [1.33–5.78]), second pregnancy trimester (OR = 6.12, 95% CI [1.80–20.85]) and presence of BV flora (OR = 8.73, 95% CI [0.50–153.60]) were associated with higher rates of *G. vaginalis* colonization in these pregnant women.

## DISCUSSION

This is the first epidemiological study conducted in Portugal with the aim to assess the prevalence of BV among pregnant women, as well as to identify risk factors for BV and *G. vaginalis* colonization during pregnancy. In contrast with the high frequency of *G. vaginalis*, BV was diagnosed in only 3.88% of studied population. This BV prevalence rate is much lower than that reported by other Portuguese studies (*Guerreiro, Gigante & Teles, 1998*;
**Table 4  Characterization of women with or without vaginal colonization by *G. vaginalis*.** Sociodemographic, medical, reproductive, behavioral and microbiological variables among women with or without *G. vaginalis* colonization.

| Variables | GV positive ($n = 139$) | GV negative ($n = 67$) | *p* value | OR | 95% CI |
|---|---|---|---|---|---|
| Sociodemographic | | | | | |
| Age (mean ± SD, years) | 29.94 ± 5.35 | 30.12 ± 4.78 | 0.88 | | |
| <30 years | 59 | 27 | | 1.09 | 0.60–1.98 |
| ≥30 years | 80 | 40 | | 0.92 | 0.51–1.66 |
| Educational level | | | 0.02 | | |
| ≤Basic | 49 | 11 | | 2.77 | 1.33–5.78 |
| Secondary | 47 | 26 | | 0.81 | 0.44–1.47 |
| ≥University | 43 | 30 | | 0.55 | 0.30–1.01 |
| Medical | | | | | |
| Previous BV | 13 | 3 | 0.28 | 2.20 | 0.61–8.01 |
| History of chronic disease | 24 | 6 | 0.14 | 2.12 | 0.82–5.47 |
| Reproductive | | | | | |
| Previous delivery | 62 | 24 | 0.29 | 1.44 | 0.79–2.63 |
| Previous preterm delivery | 9 | 5 | 0.77 | 0.86 | 0.28–2.67 |
| Pregnancy trimester | | | <0.01 | | |
| First | 17 | 7 | | 1.19 | 0.47–3.04 |
| Second | 31 | 3 | | 6.12 | 1.80–20.85 |
| Third | 91 | 57 | | 0.33 | 0.16–0.71 |
| Behavioral | | | | | |
| Tobacco consumption | 22 | 4 | 0.07 | 2.96 | 0.98–8.98 |
| Use of intimate hygiene products | 45 | 14 | 0.10 | 1.81 | 0.91–3.61 |
| Vitamin supplementation | 124 | 54 | 0.13 | 1.99 | 0.89–4.47 |
| Microbiological | | | 0.02 | | |
| Normal flora | 94 | 56 | | 0.41 | 0.20–0.86 |
| Intermediate flora | 37 | 11 | | 1.85 | 0.87–3.90 |
| BV flora | 8 | 0 | | 8.73 | 0.50–153.60 |

**Notes.**
GV, *Gardnerella* vaginalis; OR, odds ratio; CI, confidence interval; SD, standard deviation; BV, bacterial vaginosis.
Values are given as mean ± SD or number.

*Machado et al., 2015*). Indeed, *Guerreiro, Gigante & Teles (1998)* detected a BV prevalence of 7% among 840 contraceptive users living in Lisbon region while *Machado et al. (2015)* found a BV rate of 17.33%, among 150 young Portuguese women. Despite the lower cases of pregnant women with BV reported in this study, our results were consistent with other European studies. *Cristiano et al. (1996)* described a BV prevalence rate of 4.9% in 1,441 Italian pregnant women while Gratacós and co-workers *(1999)* found BV in 4.5% among 492 Spanish women with low risk pregnancies. Akinbiyi and colleagues *(2008)* conducted a randomized prospective study to determine the prevalence and age distribution of *Candida albicans* and BV among English pregnant women, and found a BV prevalence of 3.54% (38/1073) with the majority of BV cases belonging to the age group of 21–30 years.

Contrasting with the very low prevalence of BV, in our study population, we found a *G. vaginalis* colonization rate of 67.48%. A significant higher colonization by *G. vaginalis* not associated to BV has been reported elsewhere (*Cox et al., 2016*; *Janulaitiene et al., 2017*).

However, other epidemiological studies reported *G. vaginalis* colonization rates much lower than ours (*Pépin et al., 2011*; *Schwebke, Flynn & Rivers, 2014*; *Silva et al., 2014*). These differences might be the result of significant different populations at study. In fact, recent genomic studies described that vaginal microbiome in pregnancy is unique and distinct of the non-pregnant women (*Aagaard et al., 2012*; *Jespers et al., 2015*).

The observation that *G. vaginalis* colonization is not sufficient to cause BV, is not new (*Aroutcheva et al., 2001*; *Fredricks et al., 2007*; *Menard et al., 2008*). However, recent genomic studies have highlighted that *G. vaginalis* found in healthy women have distinct genetic profiles than isolates from women with BV (*Schellenberg et al., 2016*; *Janulaitiene et al., 2017*). Interestingly, it has been proposed that some of the known *G. vaginalis* genotypes are, in fact, distinct species (*Cerca et al., 2017*). This is supported by full genome sequence analysis and microbiology functional studies (*Harwich et al., 2010*; *Yeoman et al., 2010*; *Castro et al., 2015*).

## CONCLUSIONS

For the first time, BV and *G. vaginalis* prevalence among Portuguese pregnant women were determined. We found that BV prevalence was low but *G. vaginalis* colonization was very high. Importantly, due to the small sample size, associations between BV and potential risk factors should be made with caution and as such, further studies involving a large number of participants and in different regions of the country should be performed in the future, to confirm our observations.

## ACKNOWLEDGEMENTS

We would like to thank the medical doctors of the Obstetrics and Gynecology of Hospital of Braga and Unidade Local de Saúde de Matosinhos, namely: Joana Barros, Bárbara Ribeiro, Catarina Peixinho, Catarina Vieira, Joana Félix, Leonor Bivar, Luís Braga, for technical support in collection of vaginal samples.

### Funding

This work was supported by the FCT Strategic Project of UID/BIO/04469/2013 unit. DM was funded by the FCT individual fellowship SFRH/BD/87569/2012. NC is an investigator FCT. The funders had no role in study design, data collection and analysis, decision to publish, or preparation of the manuscript.

### Grant Disclosures

The following grant information was disclosed by the authors:
FCT Strategic Project: UID/BIO/04469/2013.
FCT individual fellowship: SFRH/BD/87569/2012.

### Competing Interests

The authors declare there are no competing interests.

## Author Contributions

- Daniela Machado performed the experiments, analyzed the data, wrote the paper, prepared figures and/or tables.
- Joana Castro performed the experiments, reviewed drafts of the paper.
- José Martinez-de-Oliveira reviewed drafts of the paper.
- Cristina Nogueira-Silva conceived and designed the experiments, reviewed drafts of the paper.
- Nuno Cerca conceived and designed the experiments, wrote the paper, reviewed drafts of the paper.

## Human Ethics

The following information was supplied relating to ethical approvals (i.e., approving body and any reference numbers):

The study was approved by the ethics committees of Unidade Local de Saúde de Matosinhos and of Hospital of Braga.

## Data Availability

The raw data has been uploaded as Data S1.

## Supplemental Information

Supplemental information for this article can be found online at http://dx.doi.org/10.7717/peerj.3750#supplemental-information.

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
