# Peer review of "Prevalence of bacterial vaginosis in Portuguese pregnant women and vaginal colonization by *Gardnerella vaginalis"

_PeerJ, doi:10.7717/peerj.3750_

## Round 0.1 · original submission · Major Revisions

Dear Authors,

The Reviewers found your manuscript to be interesting.

I would suggest you to take into consideration the Reviewers' comments and discuss or incorporate them into your manuscript in order to reach the level requested for publication.

·

Basic reporting

The Introduction sections needs more detail overview on a problem of BV in pregnancy that is not limited to the presence of G.vaginalis. References in Introduction needs to be updated (Bretelle et al., 2014; Foxman et al., 2013; Nelson et al., 2014; Kuon et al., 2017).

Experimental design

1. The authors did not mention what were exclusion criterions for study participants; please, report if any.
2. The description of PCR assay for G.vaginalis detection should be improved providing the following information i) why Staphylococcus epidermidis specific-PCR instead of human gene-specific PCR was chosen for internal control; ii) what segment of 16s rRNA coding gene was amplified using direct and reverse primers for G.vaginalis detection; iii) please, provide references for G.vaginalis primers close to their sequences; iv) what is a target sequence for PCR using independent set of primers (lines 118-119).
3. It is not clear why “multiplex PCR” is written in line 100? Please, clarify this issue.

Validity of the findings

1. Conclusions on the risk factors for BV are not well stated as they based on the statistical data calculated from extremely low number of pregnant women with BV (8 of 206). Detection of larger number of pregnant women would help to state conclusions.
2. The weakness of the paper is pure descriptive data, which could be relevant with larger number of participants, divided into subcategories based on gestational age.
3. Qualitative detection of solely G.vaginalis does not render scientific validity and relevance as has been already clearly stated that G.vaginalis is a common constituent of vaginal flora of healthy women. The paper could be improved by quantifying G.vaginalis in vaginal samples and differentiating G.vaginalis by subgroups or by accessing composition of vaginal flora by molecular methods.

Reviewer 2 ·

Basic reporting

The manuscript is clear and well written.

Please change "intimal products" at line 73 into "intimate products".

Experimental design

This is a cross-sectional study of 273 women, it should be stated more clearly that only one sample point was taken for each participant.

Please explain why (line 97) the bacterial suspension was incubated in a heating block.

Validity of the findings

Since only a very small percentage of the participants were suffering from BV, the validity of the significant association between BV and history of chronic disease can be questioned. I would be cautious stating this, also taking in mind the p-value of 0.09. Please give also more information on the meaning of "chronic disease" and the validity of this parameter, since it might a "too general condition".

An interesting addition to this work would be the characterisation of Gardnerella strains of women who had BV, and comparison with the non-BV strains in terms of biofilm development, pathogenicity.

Some findings in your discussion can be ameliorated by referring to already published literature. The relation between vaginal microbiota and smoking behavior has been described before by Brotman et al (BMC Infect Dis. 2014) and the changing vaginal microbiota in pregnant women has been described by Jespers et al. (BMC Infect Dis. 2015) and Aagaard et al. (PLoS One. 2012).

---

## Round 0.2 · accepted · Accept

The Reviewers have found that the comments they raised have been adequately addressed and the manuscript has reached the level required to be suitable for publication.

·

Basic reporting

Corrected Introduction with updated references is connected with the content of the paper.

Experimental design

Concerning PCR control: using S.epidermis as a control the authors a priori state that this bacterium persists in the vagina of each women participated in the study, as the they explained that this control "It is indifferent if the exogenous DNA is of human source or not". Human source control namely, ensures that 1) DNA was successfully extracted from the vaginal sample and 2) monitoring of the PCR inhibitors. Explanations concerning reasons for selection of S.epidermis DNA ("We used a S. epidermidis DNA control, because (i) we have plenty of that DNA source in our lab and (ii) the sequenced used is specific for S. epidermidis") have no scientific basis.

Validity of the findings

Correction performed by the authors made data statistically sound and conclusions are better stated.

Additional comments

In the reply to the reviewers the authors stated that "We choose to publish this article in PeerJ, because this journal guideline clearly states that the review process should be focused on the science, and not on the impact of the paper ". However, the PeerJ claims that "we do not allow the ‘pointless’ repetition of well known, widely accepted results". This requirement is indispensably connected with the so called 'impact'.

Reviewer 2 ·

Basic reporting

No comment

Experimental design

No comment

Validity of the findings

No comment

Additional comments

All comments and questions have been answered in this new version of the manuscript.